# The Role of Complement in Angiogenesis

**DOI:** 10.3390/antib9040067

**Published:** 2020-12-01

**Authors:** Maciej M. Markiewski, Elizabeth Daugherity, Britney Reese, Magdalena Karbowniczek

**Affiliations:** Department of Immunotherapeutics and Biotechnology, Jerry H. Hodge School of Pharmacy, Texas Tech University Health Sciences Center, Abilene, TX 79601, USA; Beth.Daugherity@ttuhsc.edu (E.D.); britney.reese@ttuhsc.edu (B.R.)

**Keywords:** complement, angiogenesis, cancer, ocular pathology

## Abstract

The link of the complement system to angiogenesis has remained circumstantial and speculative for several years. Perhaps the most clinically relevant example of possible involvement of complement in pathological neovascularization is age-related macular degeneration. Recent studies, however, provide more direct and experimental evidence that indeed the complement system regulates physiological and pathological angiogenesis in models of wound healing, retinal regeneration, age-related macular degeneration, and cancer. Interestingly, complement-dependent mechanisms involved in angiogenesis are very much context dependent, including anti- and proangiogenic functions. Here, we discuss these new developments that place complement among other important regulators of homeostatic and pathological angiogenesis, and we provide the perspective on how these newly discovered complement functions can be targeted for therapy.

## 1. Introduction

The complement system plays a key role in innate immunity against pathogens and links the initial innate immune responses with the subsequent adaptive immunity [1,2]. Large quantities of liver-derived complement proteins are always present in plasma and other body fluids as inactive precursors (zymogens), awaiting enzymatic cleavages via one of three canonical pathways: the classical, mannose-binding lectin, or alternative. In addition, the proteolytic enzymes present in inflamed tissues may cleave complement proteins and lead to activation of complement by what was termed the extrinsic pathway [3]. Through the cascade of proteolytic cleavages and protein-to-protein interactions, the protein complexes with enzymatic activity (C3 and C5 convertases), composed of large cleavage fragments, are formed on the activating surfaces [4]. They contribute to the progression and amplification of the complement cascade. The other cleavage fragments, usually smaller, interact with immune cells, including monocytes, macrophages, dendritic cells, granulocytes, NK cells, and B and T lymphocytes. In addition to within the liver, complement proteins are produced locally in immune, neuronal, epithelial, and tumor cells, fibroblasts, osteoblasts, and adipocytes [5]. Small complement cleavage fragments known as anaphylatoxins, C3a, C4a, C5a, in particular C5a, act as potent inflammatory mediators. C3b and C4b function as opsonins that flag pathogens to aid in recognition and phagocytosis by professional phagocytes. C5b-9, known as the terminal complement complex (TCC) or the membrane attack complex (MAC), forms a cytolytic or sublytic pore in the membranes of bacteria or cells. The complement system was originally described to aid (complement) antibodies in killing bacteria, thus the name “complement.” However, recent studies have revealed that complement is involved in several immune and inflammatory processes triggered by infection or non-infectious pathologies and contributes to several physiological and pathological process that are not strictly linked to the fight against microbes [6]. This review focuses on the non-immune complement functions associated with angiogenesis.

## 2. Vasculogenesis and Angiogenesis in Health

New blood vessels are formed through two functionally distinct processes termed vasculogenesis and angiogenesis [7]. *Vasculogenesis* is the formation of the entirely new vasculature during early embryonic development from precursor cells and is pivotal for the development of the cardiovascular system. In this process, mesodermal cells expressing vascular endothelial growth factor receptor 2 (VEGFR-2) are stimulated in a paracrine fashion by VEGF released from endoderm [8]. This stimulation contributes to the conversion of multipotent mesodermal cells to angioblasts or endothelial precursor cells that are located at the periphery of blood islands that emerge from the mesoderm as the earliest vascular structures in embryo. The fusion of the multiple blood islands leads to the formation of the primitive capillary network, which, upon the initiation of blood circulation, converts into the arteriovenous vascular system (Figure 1) [9]. The growth and maturation of this primitive vascular system requires the tight coordination of cell proliferation, differentiation, migration, matrix adhesion, and changes in cell signaling. Although vasculogenesis was initially thought to be exclusively linked to the development of the cardiovascular system in embryo, recent evidence indicates that a similar process may contribute to the creation of new blood vessels in ischemic tissues and tumors. This process has been termed postnatal vasculogenesis [10]. In contrast to de novo formation of blood vessels through vasculogenesis, *angiogenesis* refers to the growth of new capillaries from the existing vasculature (Figure 1) [11]. In adult organisms, endothelial cells remain mostly in the quiescent state and do not proliferate. Therefore, angiogenesis is rarely seen under normal physiological conditions except placental angiogenesis, angiogenesis in ovaries to transform the ovulated follicles into the corpus luteum, and in the uterus to restore the endometrium to be receptive to embryo implantation [12]. In addition, angiogenesis and vasculogenesis are required for wound healing to repair damaged vasculature and accelerate the healing process [13]. This angiogenic response is considered be a part of normal homeostatic regulation, although it occurs as a response to tissue injury. 

### Complement in Homeostatic Angiogenesis

#### Wound Healing

Normal (acute) wound healing progresses through several well-orchestrated and, to some extent, overlapping phases that encompass: clotting at the site of injury, an acute inflammatory response and associated recruitment of inflammatory and interstitial cells, the subsequent proliferation of recruited or resident cells, growth of capillaries, extracellular matrix protein deposition, regeneration of parenchymal and non-parenchymal cells, and tissue remodeling [13]. The blood vessel grows within aptly-named granulation tissue, which fills the wound during the repair process. This new vasculature is required to repair damage to the blood vessels and provide nutrients and oxygen to the highly metabolically-active cells in the sites of tissue repair. Both VEGF-mediated sprouting angiogenesis (SA) and vasculogenesis contribute to this process. Bone marrow-derived endothelial progenitors are involved in de novo blood vessel formation through vasculogenesis [13]. Complement effectors generated through complement activation, especially anaphylatoxins, are some of the most potent inflammatory mediators in humans [3]. Therefore, their involvement in the acute inflammatory reactions in wound repair was highly anticipated. 

However, in contrast to this anticipation, a recent work demonstrated that complement protein C1q is involved in wound healing vascular changes, independently of complement activation [15]. Although C1q is implicated in the initiation of the classical pathway of complement activation, it also directly regulates function of immune and non-immune cells through binding to the C1q receptors that are expressed on these cells [16], and such interaction of C1q with its reciprocal receptors on endothelial cells may be involved in C1q-mediated angiogenesis. This notion was based on C1q-positive immunohistochemistry staining on endothelial and inflammatory cells of granulation tissue in the healing wound, in the absence of C4 and C3, which are required for the complement activation cascade to proceed [15]. The C1q found in the wounds was likely derived from plasma. However, local generation of C1q was also demonstrated in this study by quantitative PCR and in situ hybridization [15]. In vitro assays with cultured endothelial cells demonstrated the impact of the globular C1q head on the permeability of the endothelial monolayer and proliferation and migration of endothelial cells. Importantly, endothelial cells migrated toward a C1q gradient, suggesting a role of C1q in the tip formation during SA, which is similar to the role of VEGF in this process [15]. Like VEGF, C1q promoted tube formation in the tube formation assay. These proangiogenic C1q properties were further corroborated by the impact of C1q on vessel sprouting in an ex vivo rat aortic ring assay. Finally, wounds from C1q-deficient mice were less vascular and closed more slowly than in wild-type controls [15]. In contrast, C3 deficiency did not impact wound vasculature suggesting that the canonical complement activation pathways do not play a major role in wound angiogenesis. Adding back C1q to C1q-deficient mice entirely abrogated the effects of C1q deficiency on wound angiogenesis. Topical application of C1q in a rat model of wound healing accelerated the healing process and increased vascularity of the wounds. Overall, this work has demonstrated C1q proangiogenic functions that are independent from complement activation [15]. 

In contrast, a different study demonstrated that experimental wounds actually heal better in C3, C5 and C5a receptor 1 (C5aR1)-deficient mice than in wild-type controls. The wounds from C3-deficent mice were enriched in mast cells and had better developed vascularity than wounds in wild-type controls. The inflammatory infiltrate of wounds was reduced in C3-deficient mice [17]. Taken together, both studies indicate that properly balanced complement activity is required for smooth wound healing. C1q-mediated angiogenesis seems to be essential for proper wound healing. However, excessive inflammatory infiltrate linked to complement C5a may delay angiogenesis and healing. 

## 3. Vasculogenesis and Angiogenesis in Disease

In contrast to these examples of “beneficial” homeostatic angiogenesis, dysregulated angiogenesis or vascular dysfunction contributes to several common and life-threatening diseases including cancer and diabetes mellitus. In addition to diabetic retinopathy, several other ocular pathologies are associated with aberrant vasculature and angiogenesis. In general, the angiogenic switch is triggered by ischemia followed by hypoxia, as is the case in cancer and other conditions, in which metabolic oxygen demand is higher than the available supply. 

### 3.1. Complement in Pathological Angiogenesis 

#### 3.1.1. Cancer

In contrast to normal tissues, exponentially growing tumor mass requires dense and constantly growing vasculature to meet its metabolic demands. Therefore, neoangiogenesis is essential to support tumor growth [18]. Even with robust angiogenesis permanently switched on in tumors, the tumor microenvironment remains hypoxic, and the central portions of solid tumors often become necrotic due lack of sufficient blood and oxygen supply. Hypoxia-induced changes in the tumor microenvironment, soluble factors secreted from tumor cells, and interplay between the resident and recruited components of this microenvironment all contribute to SA in tumors [19]. SA is initiated by several proangiogenic growth factors, including VEGF, produced in the hypoxic and nutrient-deprived tumor microenvironment. Thus, VEGF stimulated endothelial cells leave their quiescent state and proliferate. Matrix metalloproteinases (MMPs) released from activated endothelial or infiltrating cells digest the basement membranes and interstitial stroma allowing endothelial cells to migrate. Migrating frontrunner endothelial cells form a tip that follows the gradient of proangiogenic factors. Tip endothelial cells have low proliferative capacity. However, they are followed by so-called stalk cells, which proliferate and cause the sprout elongation. The vascular lumen is formed between two parallel layers of stalk cells while the sprout elongates (Figure 1). Once tip cells from neighboring sprouts meet, they anastomose to form a new perfused blood vessel. The perfusion causes the maturation and stabilization of these new blood vessels that involves deposition of basement membrane components and establishing pericyte coverage [12]. Paradoxically, VEGF-triggered SA does not lead to well-developed and mature vasculature in tumors. Newly formed tumor blood vessels are convoluted, disorganized and do not have a well-developed endothelial cell lining. Furthermore, pericyte coverage of these newly formed blood vessels is limited. As a result, the vasculature of tumors is leaky and inefficient [20]. To circumvent deficiencies associated with “ineffective” SA, the invading tumor border engulfs normal blood vessels surrounding malignant tissue and incorporates these mature blood vessels into the tumor vasculature. As expected, these co-opted host vessels are functionally different from blood vessels formed through SA. For example, the endothelial cells of co-opted vessels exhibit little proliferative potential. The co-opted vessels, in addition to supplying blood to the tumor, provide the port of entry to the circulation for tumor cells that later give rise to hematogenous metastasis. Therefore, tumor invasion of normal blood vessels in close proximity worsens prognosis [21]. Interestingly, tumor cells can also function as endothelial cells forming vascular channels through the process of vascular mimicry [22]. In addition to a pivotal role of angiogenesis and vascular aberrations in primary tumor sites, the angiogenic switch is required for the progression of small avascular micrometastases to clinically overt rapidly growing lesions, and thus, promotes the wakening of tumor cells from dormancy [23]. The role of complement in cancer has been extensively studied and reviewed in the last decade [24,25,26]. Overall, it is generally accepted that complement-mediated inflammation supports immune escape of tumors through activation and recruitment of different components of the tumor microenvironment [5]. 

##### Primary Tumors

The role of complement in cancer has been postulated for decades, since complement deposition was observed in several common human solid tumors [5,27]. The presence of complement deposits in tumors was seen as an indicator of complement-mediated tumor immune surveillance. The TCC can lyse invading microorganisms through complement-dependent cytotoxicity, and therefore, the TCC was thought to contribute to complement-mediated tumor cell killing. However, more than a decade ago, the protumor functions of complement were discovered [28]. This study linked these tumor-promoting functions to suppression of antitumor immunity through the engagement of C5a with C5aR1 expressed on myeloid-derived suppressor cells (MDSC). Several follow-up studies confirmed these initial findings and discovered new complement-mediated mechanisms contributing to cancer progression. Given the proangiogenic functions of MDSC [29], it is surprising that only a few reports have linked complement to tumor angiogenesis, and in fact, none of them have demonstrated contribution of C5aR1 signaling in these cells to angiogenesis in primary sites. 

Perhaps the first study on the role of complement in tumor angiogenesis was the work demonstrating significantly reduced tumor growth and reduced vascularization of tumors from C3 and C5aR1-deficient mice in a transgenic model of ovarian carcinoma [30]. The complete complement deficiencies with both copies of complement gene lost (homozygous mice) had greater impact on tumor growth and vascularization than the partial complement deficiencies in heterozygous mice. The tumor growth and vascular density seemed to be independent from inflammatory infiltrate of tumors, but in ex vivo stimulated immune cells, these correlated with reduced cytokine production including IL-12 and IL-10 produced in macrophages, IL-10 in B cells, and IFN-γ in T cells. In vitro, C5a promoted tube formation (in the tube formation assay) to a similar extent as the VEGF isoforms, which are designated by the number of amino acids in the spliced polypeptide, and this effect was mediated by C5aR1, as the blockade of this receptor by the selective C5aR1 antagonist, PMX53 [31], abrogated this effect. Interestingly, C5a stimulation of human endothelial cells led to increased expression of VEGF_165_ isoform and C5aR1 blockade by PMX53 attenuated tube formation mediated by VEGF, suggesting that the interplay between C5aR1 and VEGF plays a role in complement-dependent regulation of tumor angiogenesis [30]. However, the potential links between VEGF and C5aR1 signaling require further in-depth investigations.

Consistent with the role of C1q in wound healing [15], C1q endothelial positivity by immunohistochemistry was found in histological specimens from melanoma and colon, lung, breast, and pancreatic carcinomas [32]. Conversely, C1s, C4, and C3 were much less abundant or absent in the same specimens, suggesting that C1q deposits were not associated with the activation of the complement cascade. C1q staining was limited to stroma, endothelial cells, and infiltrating leukocytes [32]. A similar pattern of staining was observed in liver metastases of colon carcinoma. In contrast to colon tumors, benign mucosa did not exhibit C1q staining [32]. However, the results of this immunohistochemistry study require cautious interpretation, given that conclusions are based on analysis of only six specimens per cancer. In addition, the analysis does not include adjustments for different histopathological subtypes, grade, and other clinicopathological variables. Furthermore, publicly available data from The Human Protein Atlas indicates C3-positive staining in several common solid tumors (https://www.proteinatlas.org/). In the same study [32], tumor growth was reduced and mouse survival increased in C1q-deficient mice in a syngeneic (B16) model of melanoma. Surprisingly, the authors did not observe the impact of C3 deficiency on the growth of B16 tumors, which was inconsistent with reports documenting the role of C3 in the same murine model of melanoma [33]. Immunohistochemistry of mouse tumors was similar to data from human cancers. The study in bone marrow chimeras suggested that C1q from non-bone marrow-derived cells contributes to the tumor growth in this model. The reduced tumor growth in C1q-deficient mice correlated with reduced vascular density and reduced lung metastatic burden ([32] and Figure 2). Consistently, in a syngeneic model of model of HPV-induced cancer (TC-1), C1q deficiency was linked to morphologically aberrant vasculature in tumors characterized by shorter and disorganized vessels. In contrast, tumor blood vessels from C4-deficienct mice were normal in the same study [34]. In tumor sections obtained from the large cohorts of patients with renal cell carcinoma (RCC), C1q immunofluorescence was mainly detected in infiltrating cells expressing markers of macrophages. However, similar to the healing wounds [15] and specimens of melanoma and colon, lung, breast, and pancreatic carcinomas [32], C1q positivity was also observed in CD31^+^ cells, suggesting crosstalk of complement with RCC tumor vasculature [34]. However, in contrast to the previous work on C1q functions in tumors [32], C1q in RCC seems to trigger the classical pathway because C1q deposits on tumor cells co-localized with IgG and C4d, and C3 cleavage products were also detected in these tumors [34].

The contribution of complement to RCC is also evident from the most recent research, which demonstrated the striking association of several complement genes and proteins with prognosis and response to immune checkpoint inhibitors. This study has also shown reduced vascular density in C3aR-deficient and wild-type mice treated with C5aR1 and C3aR inhibitors in a mouse model of RCC. Consistently, several proangiogenic factors were downregulated in complement-deficient or wild-type mice treated with complement inhibitors [35]. 

##### Premetastatic Niche

A recent study documented the role of complement in cancer metastasis and, specifically, C5aR1-mediated recruitment of MDSC to the premetastatic niche with the impact of this recruitment on CD8^+^T cell mediated antitumor immunity and lung and liver metastatic burden [36]. The premetastatic niche represents a stage during the metastatic progression of cancer, in which tumors in primary sites induce changes in metastasis-targeted distal organs prior to the arrival of very first tumor cells. These changes are thought to be related to the factors secreted from tumors or delivered through the external vesicle cargo. However, complement was not linked to the premetastatic niche prior to the paper by Vadrevu et al. [36]. Therefore, this work added complement activation in the premetastatic niche as a source of additional mediators promoting metastasis (Figure 2). In a model of metastatic breast cancer, the deposition of C3 cleavage fragments, indicating complement activation, was found in the lungs starting from day 4 after injecting tumor cells into the mammary fat pad and before the arrival of tumor cells to this organ. Significant inflammatory infiltration and thickening of intra-alveolar septa reflecting pneumonitis, that was named premetastatic pneumonia, was also present at this stage [25,36]. The vast majority of infiltrating cells were identified as MDSC and their recruitment was mediated by C5a/C5aR1 [36]. Because MDSC are proangiogenic in primary tumor sites [29,37], it is conceivable that they may actually initiate angiogenesis in the premetastatic niche. Although vascular changes such as increased permeability are well-defined for the premetastatic niche [38], in general, angiogenic switching is thought to be limited to the progression of dormant micrometastases to clinically overt lesions and not triggered prior to the arrival of tumor cells [39,40]. Despite several proangiogenic factors and cells being implicated in the premetastatic niche [38,41,42], the studies documenting “true” angiogenesis at this stage of cancer are limited [38]. The transcriptomic analysis of the mouse lungs, at the stages of premetastatic niche, confirmed upregulation of pathways associated with activation of myeloid cells that can be potentially linked to angiogenesis [43]. Transcriptomic data were corroborated by gene expression profiling showing upregulation of the well-established regulators of SA including different forms of *Vegf*, *Vegfr1* and *Vegfr2*, and *Tek* genes [43]. Tek, for example, plays a key role in tip formation during SA [44]. Other factors that are linked to SA, inflammation, and changes in vascular permeability included upregulation of *IL-1b*, *Tnf* and *Mmp9* genes [43]. Importantly, IL-1β and TNF are important in activating and recruiting myeloid cells to tumors. These cells regulate angiogenesis in an MMP9-dependent fashion [45]. The upregulation of proangiogenic factors correlated well with key morphological hallmarks of angiogenesis such as increased vascular density in the interalveolar septa of the premetastatic lungs, endothelial cell proliferation, and increased vascular permeability. The increased vascular permeability was caused by reduced pericyte coverage [43]. The activation of endothelial cells was further confirmed by upregulation of TGF-β1 accessory receptor (CD105), known also as endoglin [43], because endoglin protein is rarely expressed on the quiescent endothelial cells and is a marker of tumor angiogenesis [46]. A causative role of MDSC in premetastatic angiogenesis was supported by the MDSC-depletion studies which showed a reduction in vascular density in the premetastatic lungs because of this depletion. Blockade of C5aR1 with PMX53 or genetic C5aR1 deficiency resulted in reduced vascular density in the lung premetastatic niche and was associated with overall downregulation of proangiogenic pathways, therefore implicating C5a/C5aR1 in regulation of premetastatic angiogenesis. As anticipated, reduced angiogenesis in C5aR1 deficiency or blockade correlated with reduced infiltration of the lungs by MDSC [43]. Overall, this study provides (1) preliminary evidence for angiogenesis in the premetastatic niches, which may explain a lack of dormancy in some aggressive malignancies that manifest with rapid development of metastasis even when the primary tumor site cannot be identified [47], and (2) implicates complement in premetastatic angiogenesis through the regulation of MDSC (Figure 2). Thus, the complement-MDSC crosstalk is important for both suppression of antitumor CD8^+^ T cells [36] and angiogenesis in the premetastatic niche [43]. 

#### 3.1.2. Ocular Pathologies

##### Age-Related Macular Degeneration

Age-related macular degeneration (AMD) is the leading etiology of irreversible vision loss in the elderly, which is caused by the progressive degeneration of the photoreceptor cells in the center of the retina, known as the macula [48]. The morphological hallmark of disease is the accumulation of waste products of retinal metabolism known as drusen underneath the retinal pigmented epithelium (RPE), which is adjacent to photoreceptors and provides structural and functional support for these photoreceptors (Figure 3). RPE sits on the basement membrane that separates these cells from capillaries in the choroid, a vascular layer in the eye There are two major forms of AMD: (1) neovascular (wet AMD, 10–15%) and (2) geographic atrophy (dry AMD, 85–90%) [49,50]. In wet AMD, which may start suddenly and progresses rapidly, the abnormal blood vessels sprout from the choroid into the retina as a consequence of VEGF secretion from RPE (choroidal neovascularization) [51]. These blood vessels are fragile and leaky. When they rupture, the resulting hemorrhage into the retina can cause blindness (Figure 3). The importance of VEGF-dependent neovascularization for the pathophysiology of wet AMD triggered the approval of anti-VEGF agents for treatment of this condition. After 2 years of treatment with an anti-VEGF agent, more than 95% of patients can expect to remain within 3 lines of their baseline visual acuity on a standard eye testing chart, and up to 40% can expect an improvement of 3 lines in the affected eye over baseline [52,53]. Dry AMD progresses slowly and causes some degree of visual impairment including, sometimes, complete blindness (Figure 3). However, the magnitude of visual impairment is lower in dry than in wet AMD which progresses inevitably and rapidly to blindness when left untreated [48]. Both forms of AMD appear to be linked to the proinflammatory factors that are produced by stressed RPE or inflammatory cells, including macrophages, infiltrating the retina [50]. In both patients and mouse models, early deposition of complement components C3 and C5 is observed in retina, suggesting a contributing role of complement in AMD (Figure 3). The presence of C3a and C5a in drusen of patients with AMD and in animal models of AMD further points to the pivotal role of complement activation in AMD pathogenesis [54,55]. Because C3a and C5a, but in particular C5a, are potent inflammatory mediators that activate and recruit leukocytes to inflammatory sites [3], it is likely that these mediators enhance the proinflammatory and proangiogenic environment in the retina of patients with AMD. Further evidence for the role of complement in AMD is the strong associations of several complement genetic variants with an enhanced risk of AMD, as demonstrated by large family-based studies and a genome-wide association study. This work led to the discovery of the complement factor H gene variant (*CFH*; Tyr402His) that is strongly associated with higher AMD prevalence [56]. Furthermore, common genetic variants in or near complement genes *CFH*, *C2*, *CFB*, *C3* and *CFI* explain almost 40–60% of the heritability of AMD [57]. The remaining heritable AMD cases could possibly be linked to the rare genetic variants. Several of these variants are linked to complement genes including *CFH* and *CFI* [49]. The functional significance of genetic alterations in complement genes for mechanisms behind AMD remain to be established. However, since risk of AMD is often associated with genetic variants of genes for complement regulators (*CFH* and *CFI*) and, at least, some of these variants associate with impaired function of these regulators, it is conceivable that the excessive complement activation contributes to AMD. This notion is consistent with the concept of a harmful role of the highly proinflammatory retinal milieu in AMD.

Thus, restoring factor H function and, consequently, inhibiting the alternative pathway may offer a new therapeutic avenue for AMD. A study utilizing a mouse model of AMD-laser induced choroidal neovascularization (CNV) explored the impact of the deficiency in complement factor B, which is required for the alternative pathway, on vascular changes. In addition, the effect of treatment with a recombinant complement receptor 2 (CR2)-*N*-terminal factor H domain construct (CR2-FH) on vasculature was tested in the same model. The *N*-terminus of mouse factor H contains the alternative pathway inhibitory domain whereas CR2 was used to target the factor H fragment to complement cleavage fragments. After induction of choroidal neovascularization by laser photocoagulation, mice were treated with CR2-FH, which significantly reduced CNV to the extent observed in factor B-deficient mice. Reduced CNV correlated with reduced damage to the retina [58]. 

##### Proliferative Retinopathies

Proliferative retinopathies are often associated with neovascularization of the retina resulting from vascular dysfunction, vessel loss, and hypoxia. Newly formed blood vessels are, however, dysfunctional and leaky, and therefore, they lead to impairment and, eventually, vision loss (Figure 3). These vascular alterations contribute to pathogenesis of common retinopathies, including diabetic retinopathy and retinopathy of prematurity, which all cause impaired vision in millions of patients in the USA [59,60]. From 2010 to 2050, the number of Americans with diabetic retinopathy is expected to nearly double from 7.7 million to 14.6 million according to Center for Disease Control and Prevention. In contrast to its harmful role in a model of AMD, the alternative pathway of complement activation appears to prune neovessels in oxygen-induced retinopathy (OIR) mouse models, thereby reducing severity of disease [60]. Factor B-deficient mice had a significantly higher number of neovessels than wild-type controls in the OIR model, and these neovessels persisted longer in these mice during the resolution phase, in which vasculature normalizes. However, eventually neovessels disappeared in these mice, indicating that the alternative pathway is not the only player in the process of retinal neovascularization [60]. Although the number of neovessels was higher in retinas of factor B-deficient mice, the proliferation of endothelial cells remained at the level of wild-type controls, indicating that the removal of neovessels is affected in these mice rather than neovessel formation [60]. These findings are consistent with the similar expression of *Vegf* isoforms and *Vegfr2*, which are key factors inducing endothelial cell proliferation, despite the loss of factor B [60]. Furthermore, immunofluorescence in wild-type mice demonstrated deposition of factor B in neovessels but not in normal vasculature, indicating that neovessels specifically induce complement activation. Factor B deposits were present during neovessel formation and the resolution phase and correlated with apoptosis of endothelial cells. The reduced expression of complement regulator CD55, known as decay accelerating factor, which inhibits both the classical and alternative pathway C3 convertases, appears to render neovessels vulnerable to complement-mediated removal [60]. 

Like factor B knockouts in the OIR model, C3-deficient mice displayed increased neovascularization in the model of retinopathy of prematurity (ROP) and in an in vivo matrix plug assay [59]. In addition, C5 neutralization by monoclonal antibody, C5aR1 blockade by PMX53, and C5aR1 deficiency led to increased vascularization of retina in ROP [59]. The mechanisms of complement-mediated inhibition of pathological angiogenesis involved regulation of macrophages through C5aR1 signaling. Stimulation of this receptor in macrophages induced antiangiogenic phenotypes in these cells that were associated with the increased expression and secretion of antiangiogenic soluble VEGFR1. Consistent with these findings, macrophage depletion reversed phenotypes associated with complement deficiencies [59].

## 4. Targeting Complement-Mediated Angiogenesis

Complement-dependent vascular alterations discussed here contribute to the most common human diseases that are associated with significant morbidity and mortality. Diabetes mellitus with subsequent cardiovascular complications and cancer are leading causes of mortality world-wide [61]. Despite the preclinical evidence of complement involvement in these pathologies and availability of therapeutics targeting complement, complement-based therapies for these common diseases have not entered the clinic [62]. However, some complement-based therapies were or are currently being tested in clinical trials. Perhaps, the most tested application for complement therapeutics is in AMD. Therapeutics targeting C3, Factor D, Properdin, C5 and CD59 were or are being tested in clinical trials for AMD [62,63]. Moreover, a C5aR1 antibody (IPH5401) entered a phase I clinical trial in combination with anti-PDL1, durvulumab, in patients with selected advanced solid tumors (NCT03665129). The expected study completion is June 2021. These recent developments have promise for more complement-based therapies for common human diseases. The limited progress in targeting complement-mediated vascular aberrations in the clinic can be associated with a relative paucity of studies documenting the role of complement in pathological angiogenesis, some conflicting data on the pro- vs. antiangiogenic role of complement, and limited interest of industry in targeting complement pathways for therapy. In cancer, the additional hurdles include uncertainty as to which of the complement-dependent mechanisms of cancer progression should be specifically targeted for therapy and determining which cancer patients will benefit most from complement-based interventions [35].

## Figures and Tables

**Figure 1 antibodies-09-00067-f001:**
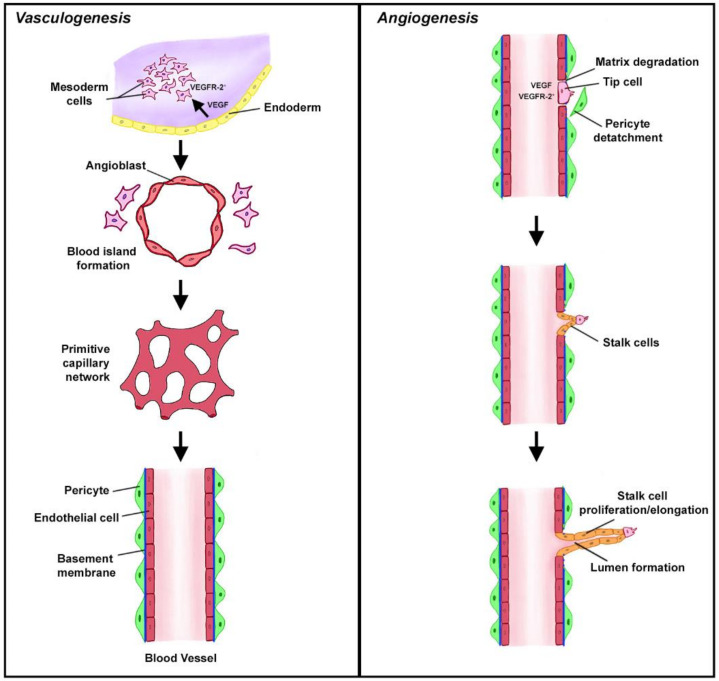
Vasculogenesis vs. angiogenesis. Vasculogenesis in embryos (**left**) is the de novo creation of new vasculature from primitive mesodermal cells that express VEGFR2. VEGF released from neighboring endodermal cells converts these VEGFR2^+^ cells into angioblasts surrounding the blood islands. These islands fuse to form the primitive vascular network that eventually converts into the arteriovenous system. In contrast, angiogenesis (**right**) is a formation of new vessels from the existing vasculature. The initial steps involve degradation of the extracellular matrix (ECM), detachment of pericytes, and tip formation. The non-proliferating tip endothelial cells move toward proangiogenic factors and are followed by dividing stalk cells. These drawings are based on the figures included in [12,14].

**Figure 2 antibodies-09-00067-f002:**
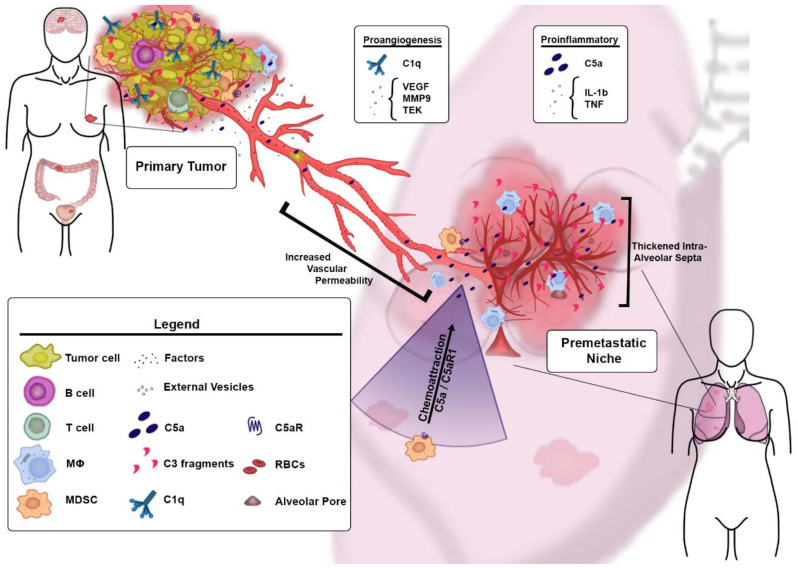
Overview of the role of complement in cancer-associated angiogenesis. C1q, C3 and C5aR1 were implicated in angiogenesis in primary tumors in cancer models and patients. In the premetastatic niches, angiogenesis seems to be regulated by MDSC activated and recruited to these niches through C5aR1-mediated signaling. MDSC secrete proangiogenic and proinflammatory factors contributing to this process.

**Figure 3 antibodies-09-00067-f003:**
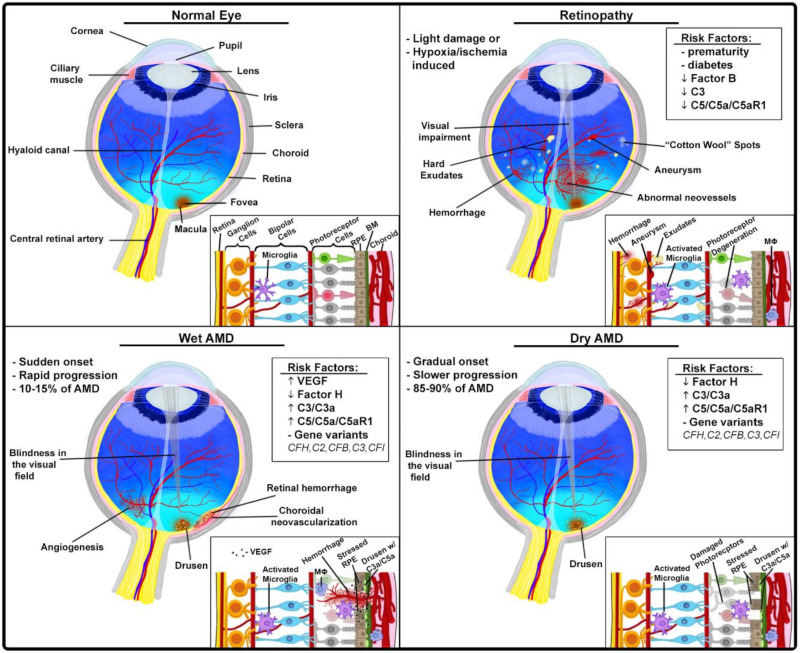
Ocular pathologies with complement involvement. Anatomy and histology of normal eye (**upper left**); proliferative retinopathies (**upper right**): the process of neovascularization leads to formation of blood vessels that are fragile and leaky, increased permeability or rupture of newly formed blood vessels and subsequent hemorrhage cause impairment or loss of vision, in mouse models of these retinopathies, complement seems to have a protective role; wet AMD (**lower left**) and dry AMD (**lower right**): in both forms, excessive complement activation and subsequent generation of complement effectors (C3a and C5a) has been implicated; frequently AMD is linked to *FH* polymorphism; in wet AMD, VEGF-driven neovascularization, possibly also regulated by complement, leads to severe impairment or loss of vision.

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
