# Peer review of "The Role of Complement in Angiogenesis"

_2073-4468, 2020, doi:10.3390/antib9040067_

Round 1

Reviewer 1 Report

This review focus on the complement involvement in blood vessel formation during health and disease. While these connections could be of interest, this review suffers from the lack of coherence between the different parts. As is presented it stands out as fragmented parts, with repetitions in content as well as language. There is a lack of a clear disposition and logical presentation. The review without summary/conclusions. 

The manuscript should be rearranged in a more logistic way. The lack of coherence is partly because complement and angiogenesis in separate parts. This also results in repetitive sections. One example is sections 2 and 4, both including AMD and complement. The sections also presents health and disease conditions - first with focus on angiogensis, subsequently from the view of complement.  One better approach would be to merge the parts, to avoid repetitions. Repetitions is also found to sentences; for instance on quite general terms when addressing complement (page 1, line 34/35) is similarly expressed in later sections page 4, line 143/144. 

Some statements lack references (example: page 7, line 242-244), thus the authors should check this thoroughly. The authors should also check for plagiarism, since some of the parts sound familiar to existing litterature. 

This review needs major revision paying attention particularly to merge the different parts so the major focus of complements role in various aspects of angiogenesis becomes more clear

Author Response

Comment: The manuscript should be rearranged in a more logistic way.

Response: We reorganized the manuscript according to the editor’s suggestions and the sections were consolidated to omit redundancies. 

Comment: Some statements lack references (example: page 7, line 242-244), thus the authors should check this thoroughly. The authors should also check for plagiarism, since some of the parts sound familiar to existing literature.

Response: Reference citations were added, and all text was original.

Reviewer 2 Report

In the manuscript, authors have described the role of complement system in angiogenesis. The manuscript is well written with clear and concise thesis statements followed by relevant discussion. The authors have discussed the dual nature of complement system, being involved in both homeostatic and pathological angiogenesis with a focus on cancer and ocular pathologies.

The manuscript should add few minor additions:

  1. References are required for line 25, 36, and 38.
  2. List of abbreviations at the end of the paper.

Author Response

Comment: References are required for line 25, 36, and 38.

Response: Reference citations were added.

Comment: List of abbreviations at the end of the paper.

Response: A list of abbreviations was added on page 10, line 401-407.

Reviewer 3 Report

The manuscript of Markiewski et al presents the role complement in angiogenesis in health and disease, including in ocular diseases, wound healing and cancer. It is a nice and timely review, with beautiful illustrations (the figures need detailed legends, though). 

Few suggestions for improvement are given below

The review will benefit from a section about pregnancy and the role of C1q in placental neovascularization. It is known that C1q deficiency is associated with impaired labyrinth development and decidual vessel remodeling and signs of pre-eclampsia. (examples of references could be: PMID: 21859968, PMID: 32374620, PMID: 32973817, PMID: 2220385, PMID: 20810993, etc).  

Line 214: 4.1. Cancer-primary tumors section: The authors may discuss a recent study PMID: 31164356, demonstrating that TC1 tumor cells, injected in C1qKO mice form smaller tumors with disorganized vascular network, while the same tumors in C4KO mice have normal vessels and also reduced tumor growth.  In part C1q impacts the neoangiogenesis via VEGFR2. Staining of C1q was also found in scarce vessels of patients with renal cancer in 3 different cohorts of patients. 

The section for ocular diseases will benefit from discussing the, at first glance, opposite findings for the role of complement in neovascularization in AND and the retinopathy of prematurity. 

Author Response

Comment: The authors may discuss a recent study PMID: 31164356, demonstrating that TC1 tumor cells, injected in C1qKO mice form smaller tumors with disorganized vascular network, while the same tumors in C4KO mice have normal vessels and also reduced tumor growth.  In part C1q impacts the neoangiogenesis via VEGFR2. Staining of C1q was also found in scarce vessels of patients with renal cancer in 3 different cohorts of patients.

Response: We added a brief discussion of this study on page 5-6, line 221 and following.

Round 2

Reviewer 3 Report

Contrary to what is stated in the reply letter, my comment seems to not have been addressed neither in line 221 nor elsewhere. 

In the paragraph of current lines 237-243 the results of renal cancer should be discussed, where C1q staining was detected in blood vessels and as deposits on tumor cells (not only in macrophages) and C3 deposits were detected in this cancer. These results are obtained in large cohorts. 

Author Response

We apologize for the omission of immunofluorescence/staining data. We discuss these data in the revised manuscript (lines 224-231).
